# IQOS^TM^ vs. e-Cigarette vs. Tobacco Cigarette: A Direct Comparison of Short-Term Effects after Overnight-Abstinence

**DOI:** 10.3390/ijerph15122902

**Published:** 2018-12-18

**Authors:** Karolien Adriaens, Dinska Van Gucht, Frank Baeyens

**Affiliations:** 1Faculty of Psychology and Educational Sciences, KU Leuven—University Leuven, Tiensestraat 102, 3000 Leuven, Belgium; dinska.vangucht@thomasmore.be (D.V.G.); frank.baeyens@kuleuven.be (F.B.); 2Applied Psychology Unit, Thomas More University of Applied Sciences, Molenstraat 8, 2018 Antwerp, Belgium

**Keywords:** tobacco harm reduction, electronic cigarettes, heat-not-burn tobacco products

## Abstract

*Introduction*: Research from Philip Morris International’s science division on its Heat-not-Burn product IQOS^TM^ focused on its chemical, toxicological, clinical, and behavioral aspects. Independent research on the experiences and behavioral aspects of using IQOS^TM^, and how it compares to e-cigarettes, is largely lacking. The current randomized, cross-over behavioral trial tried to bridge the latter gaps. *Methods*: Participants (*n* = 30) came to the lab on three consecutive days after being overnight smoking abstinent. During each session, participants used one of three products (cigarette, e-cigarette, or IQOS^TM^) for five minutes. Exhaled CO (eCO) measurements and questionnaires were repeatedly administered throughout the session. *Results*: Smoking a cigarette for five minutes resulted in a significant increase of eCO, whereas using an IQOS^TM^ resulted in a small but reliable increase (0.3 ppm). Vaping did not affect eCO. Cigarette craving reduced significantly after product use, with the decline being stronger for smoking than for e-cigarettes or IQOS^TM^. Withdrawal symptoms declined immediately after smoking or using IQOS^TM^, and with some delay after vaping. IQOS^TM^ scored higher in terms of subjective reward/satisfaction and was slightly preferred to the e-cigarette. *Discussion*: Short-term use of IQOS^TM^ has a minimal impact on eCO, is equally effective in reducing cigarette craving and withdrawal symptoms as an e-cigarette, and is slightly preferred.

## 1. Introduction

People who choose to smoke, subject themselves to one of the leading causes of years of potential life lost (YPLL) [1,2,3]. Many smokers try to quit, but traditional smoking cessation aids, or solely relying on willpower, unfortunately, only help a small minority to quit successfully and, especially, to remain smoking abstinent [4,5,6,7]. More specifically, after one particular quit-attempt only 3–5% of smokers solely relying on willpower achieve long-term (six to 12 months) smoking abstinence [7], whereas these rates double or at best triple when using Nicotine Replacement Therapy (NRT) or smoking cessation medication [5,6]. For example, in a recent analysis of the long-term outcomes of the Stop Smoking Services in England (standardly providing a combination of behavioral counseling plus NRT or smoking cessation medication), quit rates after one year were only around 8% [4]. Tobacco Harm Reduction (THR)—encouraging the substitution of low-risk alternatives—may be an alternative, more feasible way of attaining smoking cessation, especially for those smokers who cannot or do not want to cease all tobacco and/or nicotine consumption [8,9,10,11]. Alternative low-risk nicotine products include smokeless-tobacco (e.g., Swedish snus), long-term use of NRT, and electronic cigarettes (e-cigarettes) [2,11,12].

Almost a decade of extensive research on e-cigarettes has recently been evaluated and summarized in several comprehensive reviews [13,14,15]. These reviews, as well as individual studies, unanimously concluded that, beyond any doubt, using an e-cigarette (“vaping”) presents substantially less health risks than smoking; and suggested that “e-cigarette health risks are unlikely to exceed 5% of those associated with smoked tobacco products, and may well be substantially lower” [13,14,15,16,17,18]. In addition, there is convincing evidence that for many smokers, e-cigarettes are a valuable tool to assist in quitting smoking [13,14,15,19,20,21]. As reported in the majority of the well-conducted prospective observational cohort studies, the odds ratios of (self-reported) quitting (OR 2.69–7.88; 20–42% quitters) are substantially higher in smokers who self-select to use e-cigarettes in a quit-attempt compared to those who do not [22,23,24,25]. An important contributor to quit-smoking success seems to be regular (daily) and/or long-term use of efficient e-cigarettes. In the same vein, according to national (UK/France) and EU cross-sectional population data, about half (41–52%) of the current daily e-cigarette users report to have quit smoking, completely [26,27,28,29].

However, the prevalence of current e-cigarette use in the EU population (15 years and older) centers around no more than about 2% (4% in current smokers, 4% in ex-smokers, 0% in never smokers) [30]. Among the US population, prevalence of current adult e-cigarette use is estimated at about 3% [31]. Additionally, among current e-cigarette users, a considerable percentage (about 50%) use both cigarettes and e-cigarettes (“dual use”), and many of those dual users do not seem able, nor willing to completely quit smoking by switching to vaping [26,32]. From a health perspective, reducing smoking is a step in the good direction, but smoking no more than a few cigarettes per day still has significant effects on premature mortality and overall morbidity [33,34].

A partial explanation of dual use, and of the relatively low uptake of vaping by smokers, more generally, could be that vaping is not a satisfactory alternative for some smokers who want to switch to a low-risk alternative [35,36]. For example, a recent study with confirmed smokers found that 59% of the participants had tried out vaping but indicated several aspects they did not like about vaping, such as the (sensory and behavioral aspects of the) vaping experience, the material of which the e-cigarette is composed, the “chemical composition” of the e-liquids and the technical complexity of vaping [35]. This indicates that not all smokers may benefit from trying out vaping. As the THR vision implies offering (that is, developing, correctly informing about, and encouraging the use of) any tool that reduces the harms caused by smoking, it is our position that the availability of several different alternatives for smoking should be welcomed, rather than seen as a threat. This allows smokers to freely choose, try out, and find products that are sufficiently attractive and suit them best to reduce the harmful effects from smoking [9].

One other alternative could be the new Modified-Risk Tobacco Products (MRTPs), or more specifically, the Heat-not-Burn tobacco products (HnB products) developed by several tobacco companies [37,38]. Different HnB products exists, but one of the most prominent is the IQOS^TM^, developed by Philip Morris International (PMI) [38]. The IQOS^TM^ contains a holder which heats a tobacco stick (HEETS or heatsticks) below 350 °C, such that an aerosol is produced which the user can inhale; using an IQOS^TM^ closely mimics the smoking experience [38]. Based on their own research, PMI concluded (i) that no combustion occurs when the IQOS^TM^ is used, (ii) that the aerosol from the IQOS^TM^ contains substantially less (around 90%) Harmful and Potentially Harmful Constituents (HPHCs), compared to cigarettes, (iii) that this reduction in the HPHCs levels in the aerosol leads to reduced exposure and to reduced (in vitro and in vivo) toxicological effects, (iv) and that the risk of smoking-related diseases is probably reduced [38,39,40,41,42,43].

A substantial part of the research conducted by PMI included the clinical assessment of the IQOS^TM^ in humans [38]. First, three open-label, randomized, cross-over studies investigated nicotine pharmacokinetics and concluded that in smokers using the IQOS^TM^, a similar pharmacokinetic profile is obtained compared to smoking (IQOS^TM^: Cigarette ratios for maximal concentration, C_max_; and overall nicotine exposure (area under the concentration-time curve), AUC_0–last_ and AUC_0–∞_, varied between 88–104%) [44,45]. In line with this, the IQOS^TM^ appeared to be equally effective in controlling cigarette craving (urge-to-smoke measurements) as smoking cigarettes [44,45]. Second, several PMI studies investigated biomarkers of exposure to the HPHCs, as well as self-reported subjective measures (e.g., urge-to-smoke, product evaluation) in smokers switching to the IQOS^TM^, continuing smoking or quitting smoking, using randomized and controlled confinement studies in Europe and Japan [46,47,48,49]. In general, these studies included a four-week screening period, a confinement period of five days in which randomization took place, a safety follow-up period, and sometimes an ambulatory period (follow-up period of 90 days). These studies concluded that, compared to continuing smoking and already after five days, switching to the IQOS^TM^ resulted in significant reductions (34–94%) in the biomarkers of exposure to several HPHCs [46,47]. This effect was maintained upon follow-up and was comparable with reductions in biomarkers of exposure, due to complete smoking abstinence [46,47,48,49]. In addition, participants switching to the IQOS^TM^ reported similar and stable levels in the urge to smoke compared to continuing smoking. Product evaluations of the IQOS^TM^ and cigarettes differed across studies, but a longitudinal study showed that from day 1 to 30, the cigarette was rated better, whereas from day 30 onwards, the IQOS^TM^ achieved similar scores [49].

A recent systematic literature review confirmed that most of the research involving HnB products, so far, has been conducted by tobacco companies, such as PMI and British American Tobacco (BAT), themselves (20 out of the 31 included studies), and highlighted that the few independent research conducted, mainly focused on the chemical aspects of the HnB aerosol [50]. A comparison of results is complicated by the heterogeneity of the HnB products studied and by the different measurement methods used [50]. Nevertheless, the overall picture that emerges from independent aerosol studies largely confirmed PMI’s findings, with respect to the levels of nicotine, carbon monoxide (CO), total particulate matter (TPM), and the main other HPHCs in the IQOS^TM^ aerosol (e.g., volatile organic compounds, VOCs; aldehydes; polycyclic aromatic hydrocarbons; and tobacco-specific nitrosamines, TSNAs) [50,51,52,53,54,55,56,57,58]. Independent research with respect to (in vitro and in vivo) IQOS^TM^ toxicology is rare (an exception being a study by Leigh and colleagues), whereas, human clinical and behavioral/experiential studies focusing on the IQOS^TM^ use is virtually nonexistent [50,59].

Caponnetto and colleagues were the first to independently investigate the effects of using IQOS^TM^ on exhaled CO (eCO) levels, using a randomized cross-over trial in smokers [60]. A comparison was made between the two different HnB products (IQOS^TM^ or GLO^TM^, with the latter being a HnB product similar to IQOS^TM^ but developed by BAT) and the cigarette brand regularly smoked by participants [37,60]. First, participants were trained on how to use the HnB products at screening. Next, participants were instructed to abstain from smoking for 12 h before each of the three study days, on which they could use one of the three products following a specific puffing regime [60]. Measurements were obtained at specific times throughout the sessions. The authors concluded that, unlike what is observed when smoking a cigarette, using the HnB products did not result in increased eCO levels [60]. Two limitations of this study were that no data concerning craving and subjective experiences were obtained, and that no comparisons with vaping were made. Therefore, we conducted a three-day randomized cross-over trial, focusing on the behavioral and experiential effects of the short-term use of the HnB product IQOS^TM^, versus an e-cigarette, versus a regular cigarette, in current smokers who were novice users for both IQOS^TM^ and e-cigarettes. The two main research aims of the study reported here were: (1) to investigate the effect of using an IQOS^TM^ on eCO, acute cigarette craving, withdrawal symptoms, and subjective positive and negative experiences after overnight smoking abstinence, compared to using an e-cigarette or a regular tobacco cigarette; and (2) to investigate which product (e-cigarette or IQOS^TM^) would be preferred.

## 2. Materials and Methods

### 2.1. Participants

We recruited Dutch and English speaking participants via various channels around the University of Leuven (i.e., distribution of flyers in University buildings and local newspaper shops, social media). Inclusion criteria, based on (own) previous research [60,61], were, being a smoker for at least three years, smoking at least 10 cigarettes per day (CPD), having no intention to quit smoking in the following three months, and willing/accepting to try out several less harmful alternatives. Exclusion criteria were currently using any kind of smoking cessation therapy (e.g., smoking cessation medication, NRT, counseling), ever having owned and used an e-cigarette and/or an HnB product, having tried out an e-cigarette and/or an HnB product, during the past month, being pregnant or breast feeding, and having one or more severe medical conditions (i.e., a psychiatric condition, respiratory or heart disease, drug use other than nicotine, diabetes).

A total of 46 interested individuals signed up for the intake session, of whom 34 completed all sessions. After data collection, another four participants were excluded due to not complying with the critical inclusion criteria (i.e., e-cigarette use in recent past, number of CPD substantially <10, eCO level >10 at start of laboratory session). Two participants were not excluded despite minor violations of inclusion criteria (CPD = 9; conflicting answers on two questions assessing intention to quit smoking in the next three months). The main results did not differ when excluding or including these two participants who slightly deviated from the (secondary) inclusion criteria, but including them allowed to maintain the perfectly balanced design. The final sample size, thus, consisted of 30 participants.

### 2.2. Materials

Three products were used during the laboratory sessions—a regular tobacco cigarette, an e-cigarette and the IQOS^TM^ HnB tobacco product. Participants were asked to bring their habitual tobacco cigarettes, so that they could use one in the corresponding session (see Section 2.3. Study Design and Procedure).

During the session in which participants were asked to use the e-cigarette or IQOS^TM^, we provided each participant with the same product. We used an Eleaf iStick Power 5000 mAh battery, fixed at 8 W, with an Aspire Nautilus 2 tank containing a 1.6 Ohm coil. The e-liquid (“Base Aurora”) contained 18 mg/mL nicotine, a PG/VG ratio of 70/30, to which either a tobacco flavor (“7 Leaves”, 3 vol%) or a menthol flavor (“Mild Winter-Peppermint”, 3 vol%) was added. Both base liquid and flavors were purchased online (https://www.clubderdampfer.de and https://flavourart.com, respectively). Clearomizer type and wattage settings, as well as nicotine concentrations, were based on the results of a study by Farsalinos and colleagues, in which they found that these settings can—conditional on longer puff duration—deliver similar nicotine levels to the aerosol (0.82 mg/12 puffs at 2 s vs. 1.84 mg/12 puffs at 4 s) as combustible tobacco cigarettes (1.99 mg/12 puffs at 2 s) and slightly more than the IQOS^TM^ HnB product (1.40 mg/12 puffs at 2 s vs. 1.41 mg/12 puffs at 4 s) [53].

The IQOS^TM^ was purchased in an official IQOS-shop in the Netherlands, since HnB products are not available in the Belgian market. Both regular and menthol-flavored heat-sticks were purchased. The menthol e-liquid and heat-sticks were only destined for regular menthol cigarette smokers; however, because nobody happened to smoke menthol cigarettes, the menthol products were not used in this study.

### 2.3. Study Design and Procedure

The research protocol was approved by the Societal and Social Ethics Committee of the University of Leuven before the start of the study (G-2017 08 900) and the protocol was preregistered on aspredcited.org (#6896).

Prior to the study, interested individuals could contact the first author for more information about the study. Those willing to participate decided, through self-selection, if they were eligible to participate and subsequently subscribed for the intake session via the Experiment Management System (EMS) of the Faculty of Psychology and Educational Sciences [62].

Depending on the enrolments, intake sessions were carried out in group (with a maximum of six participants) or individually, and lasted approximately 30 min. All participants were provided with a brief explanation about the safety and practical use of the products, and the course of the study. Next, participants signed the informed consent form, filled out the intake questionnaire, and performed an eCO-measurement. At the end of the session, we scheduled the three laboratory sessions with each participant.

We used a cross-over, counterbalanced, within-subjects design for the laboratory sessions. Participants came to the lab (individually or in group, with a maximum of three participants) on three consecutive days, each time at the same hour of the day; each session lasted 70 to 80 min and followed the same procedure (see Figure 1 for all procedure details). Before each laboratory session, participants needed to abstain from smoking for 12 h. At the start of the session (T0), participants filled out questionnaires and performed an eCO-measurement. In the corresponding session, participants received a brief rehearsal on how to use the e-cigarette or IQOS^TM^. Next, participants could use one of the three products ad lib for five minutes outside the building (only one cigarette or heat-stick were allowed). In each session, only one product was used and the order of product use over the days was completely counterbalanced between participants to control for order effects. Finally, at fixed moments (T1, T2, T3, T4, and T5; see Figure 1) participants filled out questionnaires and performed eCO measurements. Participants who completed all sessions received a compensation of 50 € or, if applicable, five credits for a mandatory research course within the psychology training.

### 2.4. Outcome Measures

#### 2.4.1. Physiological Measures

eCO measurements were collected during the intake session and laboratory sessions (T0 to T5), using a piCO+ Smokerlyzer^®^ [63]. The concentration in parts per million (ppm) was noted by the researcher.

#### 2.4.2. Subjective Effect Questionnaires

Questionnaires and instructions on how to fill them out were displayed on the computer screen using Affect 5 [64]. Participants answered by clicking or entering text on the computer keyboard. The intake questionnaire assessed socio-demographics (age, gender, highest educational degree, occupation, marital status, net income per month in €, nationality—all predefined categories except for age and nationality), smoking history (‘age started smoking’, ‘age started regular smoking’, ‘how long smoking’—all open ended), current smoking behavior (‘situations when smoking’, ‘reasons why smoking’—predefined categories; ‘current average smoked CPD’, ‘most important cigarette of the day’, ‘brand/type of smoked cigarettes’—open ended), motivation to quit smoking (predefined categories), quit-smoking attempts in the past (‘number of attempts’, ‘longest quit-smoking period’—open ended; ‘quit-smoking aids used’, including those used during the longest quit-smoking period—predefined categories), experienced negative health effects of smoking (e.g., “*As a smoker I suffer from headaches*”; on Likert scales from 1 “*Never*” to 5 “*Always*”), mental health status (i.e., suffering from any psychological/psychiatric condition; predefined categories), and tobacco cigarette dependence, using the Fagerström Test for Cigarette Dependence (FTCD) [65]; see Appendix A, Table A1.

The questionnaires used during the laboratory sessions included (see Figure 1): A visual analogue scale (VAS) assessing cigarette craving, the brief Questionnaire on Smoking Urges (QSU-Brief), the Revised Minnesota Nicotine Withdrawal Scale (MNWS-R), and the modified Cigarette Evaluation Questionnaire (mCEQ, also adapted for e-cigarette and IQOS^TM^) [66,67,68,69]. The VASs were 100 mm, with on the left “*No craving at all*” and on the right “*Very strong craving*”, and were administered at each moment (T0 to T5). The QSU-Brief measures the multidimensional aspects of craving, using 10 items rated on a 7-point scale, going from “*Strongly disagree*” to “*Strongly agree*”, and was administered at T0, T1, and T5 [66]. The scale allows for the calculation of a general average score, as well as two specific factors (i.e., “*The desire and intention to smoke with an anticipation of pleasure from smoking*” and “*The relief from nicotine withdrawal or negative affect with an urgent and overwhelming desire to smoke*”). The MNWS-R measures nicotine withdrawal symptoms using fifteen items rated on a 5-point scale, going from “*None*” to “*Severe*”, and was assessed at T0, T1, and T5 [67,68]. The mCEQ assesses the reinforcing effects of smoking and contains three multi-item-domains (“*Smoking satisfaction*”, “*Psychological reward*”, “*Aversion*”) and two single-item-domains (“*Enjoyment of respiratory tract sensations*”, “*Craving reduction*”) [69]. All twelve items were scored on a 7-point scale going from “*Not at all*” to “*Extremely*”. We adjusted the questionnaire so it was also applicable for the e-cigarette and IQOS^TM^ and assessed the questionnaire solely at T1. Finally, participants were provided with some additional questions (VASs and open-ended questions) on the last day, regarding their preference for the e-cigarette or IQOS^TM^. For each theme (“*Willing to use the product for another five minutes*”, “*Willing to keep trying or start using the product*”, “*Desire/intention to go and buy the product*”, “*Willing to consider using the product to (try to) quit smoking*”) we used three VASs, one for the e-cigarette and one for the IQOS^TM^, with, for each on the left, “*Not at all*” and on the right “*Very much so*”, and one VAS that assessed the participants’ preferred choice with on the left “*E-cigarette*” and on the right “*IQOS*”. The left and right label of the latter VAS were counterbalanced across participants, and the same was true for the presentation sequence of the first two VASs. Finally, four open-ended questions assessed the weaknesses and strengths of the e-cigarette and IQOS^TM^, each compared to cigarettes; with, again, a counterbalancing across participants.

### 2.5. Statistical Analyses

We used descriptive statistics (frequencies, means, and standard deviations) to analyze the main variables assessed at Intake and the open questions (weaknesses and strengths of the products), on the last day. Before analyzing the results from the laboratory sessions, we checked the normality assumption. Normality was violated for some individual variables but due to the fact that group sizes were equal (within-subjects design), we decided to still conduct ANOVAs (because of the robustness of the statistic against violations of normality with equal sample sizes) [70]. For most variables of the laboratory sessions we carried out 3 (Condition: cigarette vs. e-cigarette vs. IQOS^TM^) × 6 (Moment: T0 to T5) ANOVAs, with subsequent planned comparisons. The additional VASs from day 3 were analyzed using *t*-tests for paired samples. For all analyses conducted, an α-level of 0.05 was used and all analyses were carried out using “Statistica 13.1” (TIBCO Software Inc., Palo Alto, CA, USA) [71].

## 3. Results

### 3.1. Participants

Participants (*n* = 30) were on average 22 years old (*SD* = 3.09) and mostly male (67%). Almost all (93%) were students with at least a high school degree (100%), having a net income of less than 1000 € (73%), and being single (73%). Almost half of the participants were of Belgian nationality (47%) with the remaining being of other nationalities (e.g., Italian, Pakistani, Indian, etc.).

Regarding smoking history, participants started smoking on average at the age of 16 (*SD* = 1.84) and started smoking regularly at the age of 18 (*SD* = 1.77). One-third had tried (*M* = 2.00 times, *SD* = 0.94) to quit smoking in the past, mainly using willpower (90%). The longest quit-smoking period (with all using willpower) had lasted on average five months (*SD* = 9.02), with a minimum of one month and a maximum of 30 months.

Currently, participants smoked, on average, 13 CPD (*SD* = 3.62), mostly filter cigarettes (79%), they were low-to-moderate cigarette dependent (*M* FTCD-score = 3.50, *SD* = 1.96), experienced rare-to-occasional negative health effects of smoking (*M* = 2.23, *SD* = 0.48), and had an average eCO level of 7.37 (*SD* = 3.39). The top three situations when they smoked included drinking alcohol (97%), being with others (93%), and being alone (87%). Reasons why participants smoked were to relax (87%), because they felt like having a cigarette (83%) and for atmosphere and sociability (80%). Only four (13%) participants reported that they currently were trying to reduce smoking. Three of them reported no intention nor concrete plans to quit smoking, and only one expressed concrete plans to quit smoking in the next three months. See Appendix A, Table A1 for all details concerning smoking history and current smoking behavior.

### 3.2. Physiological Measures

Figure 2 displays the changes in eCO levels throughout the sessions (see also Appendix A, Table A2). In each laboratory session, participants showed a significant decrease in eCO levels going from 7 ppm at Intake to 3 ppm at the start of each session (T0), *p* < 0.001, for each condition. No T0 differences in eCO levels were found between conditions, with all *p*s > 0.20. Smoking a cigarette resulted in increased eCO levels at each moment (T1–T5), compared to T0, all *p*s < 0.001. From T3 to T4, eCO levels slightly decreased, and continued doing so from T4 to T5, both *p*s < 0.001. A similar pattern was observed after using the IQOS^TM^ (all *p*s < 0.05), though in absolute terms, the increase in eCO levels after using the IQOS^TM^ was minimal (0.3 ppm). No changes occurred in the eCO levels after vaping (all *p*s > 0.06). At each moment (T1–T5), throughout the laboratory session, smoking resulted in significantly higher eCO levels compared to vaping (all *p*s < 0.001) and using the IQOS^TM^ (all *p*s < 0.001). Vaping and using the IQOS^TM^ did not lead to reliable differences in eCO levels at any moment (T1–T5; all *p*s > 0.06).

### 3.3. Subjective Effect Questionnaires

#### 3.3.1. Cigarette Craving

The changes in cigarette craving throughout the sessions, are displayed in Figure 3 (see also Appendix A, Table A3). After 12 h of smoking abstinence, at the start of each day (T0), participants reported an average cigarette craving of 66 on a scale of 100, with no differences in craving between the three conditions (all *p*s > 0.27). After using each product (T0 to T1), craving for a cigarette reduced significantly (all *p*s < 0.001), with the decline being stronger after smoking than after vaping or after using the IQOS^TM^, both *p*s < 0.001. The decline observed for the latter two did not differ, *F* < 1. From T1 to T2, only vaping resulted in a significant increase in craving, *F*(1, 29) = 8.38, *p* < 0.01. Cigarette craving significantly increased at T3, T4, and T5 for all conditions, compared to T1, with all *p*s < 0.01. At each moment (T1–T5) throughout the laboratory session, smoking resulted in a lower cigarette craving compared to vaping (all *p*s < 0.01) and using the IQOS^TM^ (all *p*s < 0.01). No differences were observed, at any moment, between using the e-cigarette and the IQOS^TM^ (T1–T5: all *p*s > 0.43).

The QSU-Brief, which was presented at T0, T1, and T5, confirmed the results obtained with the VASs for cigarette craving (see Appendix A, Table A3). More specifically, at T0 no differences were found between conditions (all *p*s > 0.10). After using each product (T0 to T1), cigarette craving was reduced (all *p*s < 0.001), with the decline being stronger after smoking than after vaping, or after using the IQOS^TM^, both *p*s < 0.05. The decline observed, after vaping, and after using the IQOS^TM^, did not differ, *F* < 1. Cigarette craving increased towards the end of each session for each condition (T1 to T5; all *p*s < 0.001), with no differences in elevation between conditions (all *p*s > 0.15). At T1 and T5, smoking resulted in lower craving scores compared to vaping (all *p*s < 0.01) and compared to using the IQOS^TM^ (all *p*s < 0.01). No differences were observed at any moment between using the e-cigarette and the IQOS^TM^ (T1 and T5, all *p*s > 0.42). Overall, the same patterns were found for the subscale “*The desire and intention to smoke with an anticipation of pleasure from smoking*”. The subscale “*The relief from nicotine withdrawal or negative affect with an urgent and overwhelming desire to smoke*”, did not show an interaction-effect, between conditions and moments, *F* < 1.

#### 3.3.2. Withdrawal Symptoms (MNWS-R)

At the start of each session (T0), participants reported relatively little withdrawal symptoms after 12 h of smoking abstinence (*M*_CIG_ = 13.93, *SE*_CIG_ = 1.32; *M_E-CIG_* = 15.20, *SE*_E-CIG_ = 1.58; and *M*_IQOS_ = 13.63, *SE*_IQOS_ = 1.42), with no differences between conditions (all *p*s > 0.36; see Appendix A, Table A4). However, smoking and using the IQOS^TM^ resulted in significant reductions in withdrawal symptoms (T0 to T1: both *p*s < 0.001), followed by a stagnation until the end of the session (T1 to T5: both *p*s > 0.10). Vaping did not result in an immediate reduction of withdrawal symptoms, *F*(1, 29) = 2.84, *p* = 0.10, but from T1 to the end of the study a decrease was observed, *F*(1, 29) = 15.31, *p* < 0.001. At T5, no differences in withdrawal symptoms were present between smoking and using the IQOS^TM^, nor between vaping and using the IQOS^TM^, both *p*s > 0.11, but vaping resulted in slightly higher withdrawal symptoms compared to smoking, *F*(1, 29) = 4.55, *p* < 0.05.

#### 3.3.3. Product Evaluation and Preferences

At T1 of each session, the mCEQ was administrated to evaluate the reinforcing effects of using the different products. For each subscale, a main effect was found of the product that was used (all *p*s < 0.05; see Figure 4 and Appendix A, Table A5). First, “*Satisfaction*” was rated the highest after smoking, followed by using the IQOS^TM^ and vaping. Smoking appeared to be more satisfying than both vaping and using the IQOS^TM^, and using the IQOS^TM^ was also more satisfying than vaping. Second, “*Psychological reward*” was for all products rated between very little and moderately, with again the cigarette being superior, followed by the IQOS^TM^ and then the e-cigarette. Third, the “*Aversion*” subscale was rated relatively low, with smoking resulting in the highest scores, similar to the scores after vaping. After using the IQOS^TM^, “*Aversion*” scores were significantly lower, compared to smoking and to vaping. Fourth, the “*Enjoyment of respiratory tract sensations*” was experienced a lot after smoking, little to moderately after using the IQOS^TM^, and very little to not at all after vaping. Lastly, the results of the “*Craving reduction*” subscale were in line with the previously reported VASs and QSU-Brief results. More specifically, smoking resulted in the highest craving reduction, followed by using the IQOS^TM^, and vaping. Using the IQOS^TM^ and vaping resulted in similar scores, with smoking showing significantly higher craving reduction scores, compared to vaping and using the IQOS^TM^.

On the individual VASs for the e-cigarette and IQOS^TM^, at the end of the laboratory sessions, participants reported only a significantly higher willingness to use the IQOS^TM^ for another five minutes compared to the e-cigarette, see Figure 5a (see also Appendix A, Table A6). For all other items no differences were found, although in absolute terms the IQOS^TM^ obtained higher scores. This pattern was also observed in the VASs, where participants indicated their preferred choice on the four items, see Figure 5b. The mean scores, indicating a preference for the IQOS^TM^, were 67.27 (*SD* = 39.23), 65.63 (*SD* = 39.90), 64.13 (*SD* = 41.28), and 69.37 (*SD* = 39.38), respectively. However, variation in product preferences was large among the participants.

Participants reported aspects they missed when using the e-cigarette and the IQOS^TM^ compared to tobacco cigarettes. Mentioned themes, including frequency of reporting by participants, were the following: (a) the taste, aroma, flavor, or smell (e-cigarette: 63% of participants; IQOS^TM^: 63%); (b) the psychophysiological effects of having used the product, such as experiencing relaxing effects (e-cigarette: 43%; IQOS^TM^: 27%); (c) the feeling/sensations of inhalation in the throat and lungs (e-cigarette: 27%; IQOS^TM^: 27%); (d) the nicotine and throat hit (e-cigarette: 23%; IQOS^TM^: 20%); and (e) the handling/gesture of smoking (e-cigarette: 17%; IQOS^TM^: 23%). Six participants (20%) reported no missing aspects for the e-cigarette and nobody (0%) did so for the IQOS^TM^. Secondly, also the strengths of both products were questioned. Themes reported included (a) better for health or less harmful (e-cigarette: 47%; IQOS^TM^: 53%); (b) the taste, aroma, flavor or smell (e-cigarette: 27%; IQOS^TM^: 17%); (c) the lack of any odor/smell after use (e-cigarette: 13%; IQOS^TM^: 20%); and (d) the psychophysiological effect (e-cigarette: 0%; IQOS^TM^: 23%). Seven participants (23%) reported no strengths of the e-cigarette and three (10%) reported no strengths of the IQOS^TM^.

## 4. Discussion

The current study, using a randomized, cross-over, counterbalanced, within-subjects design, investigated the effects of using the IQOS^TM^ HnB product compared to smoking and vaping, in overnight-abstinent regular smokers, who were novice users of e-cigarettes and HnB products. First, eCO levels decreased significantly from Intake to T0, with at T0 average eCO levels (about 3 ppm) approaching that of nonsmokers. These results seemed to confirm compliance to the abstinence rule. However, some participants spontaneously mentioned at Intake that they also did not smoke before that session, potentially explaining the relatively low eCO levels at Intake for these regular smokers. Perhaps the recruitment information was confusing concerning the abstinence rules. In addition, as observed in several other studies, our data reconfirmed that vaping does not result in any change of eCO levels [13]. As expected, after five minutes of smoking eCO levels increased significantly until T2, after which the levels slowly started to decrease again. Surprisingly, a similar pattern was observed after using the IQOS^TM^, but the increase in the eCO levels was only 11% (0.3 ppm) of the baseline values (T0 to T1), with a maximum increase of 27% (T0 to T2; 0.8 ppm), whereas, these percentages after smoking reached 135% (4.2 ppm) and 153% (4.7 ppm) of the baseline values, respectively. At first sight, these results seemed to conflict with those obtained in the study of Caponnetto and colleagues, where no significant increase was found in the eCO levels after using the two HnB products [60]. However, looking at the absolute eCO levels after the IQOS^TM^ use in the Caponnetto study, a small increase occurred up to 15 min, after using the IQOS^TM^ (maximum level during session: 4.9 ppm vs. 3.5 ppm in our study), after which the eCO levels started to decline again [60]. A plausible explanation for this nominal increase not reaching significance, is a lack of statistical power in the study of Caponnetto and colleagues—they only included 12 participants, whereas we included 30. This minor and clinically non-significant increase in the eCO levels after using the IQOS^TM^, is in line with the previously documented presence of CO in the aerosol of the IQOS^TM^, albeit at a low level of 0.53 mg/stick (which is only 1.6% of the CO in the aerosol of the 3R4F reference cigarette (32.8 mg/stick)) [38,42,60]. Namely, although no combustion takes place due to the limited heating up to 350°C when an IQOS^TM^ is used, CO may also be generated by (low-temperature) pyrolysis [72,73,74].

Second, after 12 h of smoking abstinence, participants reported moderate cigarette craving at the start of each session. Five minutes of use of each product resulted in significant decreases in cigarette craving, but smoking resulted in a craving reduction of 44%, whereas vaping and using the IQOS^TM^ only resulted in a reduction of 26% and 28%, respectively. Throughout the remaining time of the session, cigarette craving slightly increased again. These results were confirmed by the QSU-Brief questionnaire. Comparison with the studies from the PMI is difficult due to the very nature of our short-term, cross-over design, whereas PMI mainly used confinement studies to investigate the effects of the IQOS^TM^ over the course of several days of use [44,45,46,47,48,49]. In these multi-day studies, it was observed that using an IQOS^TM^ can be equally effective in controlling cigarette craving as smoking cigarettes [44,45,46,47,48,49]. Even though we did not find such pronounced effects, we likewise observed that cigarette-craving was substantially reduced after five minutes of IQOS^TM^ use, and about as much as that after five minutes of vaping, but clearly less than that after smoking.

Third, although moderate levels of cigarette craving were reported by participants in each session, little withdrawal symptoms were reported. Nevertheless, after smoking and using the IQOS^TM^ alike, withdrawal symptoms decreased immediately, while for vaping this occurred with some delay. In one of the confinement studies, Lüdicke and colleagues also found that using the IQOS^TM^ resulted in similar reductions in withdrawal symptoms, as that after smoking [48]. A possible explanation for the delay in a reduction of the withdrawal symptoms for vaping, is that the blood nicotine delivery of an e-cigarette may be slower than that of a combustible cigarette, and possibly also slower than that of an IQOS^TM^ [44,75]. In addition, participants were novice users and only had five minutes time to familiarize with the e-cigarette, which can also have resulted in less nicotine uptake and, in turn, a slower decrease of withdrawal symptoms [75]. It has been shown that longer puffs (4 s) from an e-cigarette are needed to deliver nicotine to the aerosol at levels approaching those of cigarettes, a behavior topography which probably is lacking in novice users [53]. Interestingly, in the same study, nicotine delivery to the aerosol of the IQOS^TM^ was shown not to be affected by puff duration, such that cigarette-like puffing behavior (e.g., short 2 s puffs) would not adversely affect nicotine delivery [53]. In other words, a plausible hypothesis may be that using an IQOS^TM^, but not an e-cigarette, in a smoking-like fashion (in terms of puff-topography) would result in nicotine pharmacokinetics closely mimicking those of smoking, such that a learning curve plays less of a role than for vaping.

Fourth and lastly, each product was evaluated by participants after a first-time, five-minutes-use (T1). All subscales of the mCEQ (“*Smoking satisfaction*”, “*Psychological reward*”, “*Aversion*”, “*Enjoyment of respiratory tract sensations*”, and “*Craving reduction*”) were rated lower for the IQOS^TM^ than for the tobacco cigarette. At the same time, IQOS^TM^ was evaluated better, compared to the e-cigarette concerning “*Smoking satisfaction*”, “*Psychological reward*”, and “*Enjoyment of respiratory tract sensati*ons”. These results are in line with results from PMI; more specifically, across a period of five days of confinement or across a 30-day period, all subscales of the mCEQ were also scored lower for the IQOS^TM^ compared to the cigarette (except for the Aversion-scale scores which were similar) [46,47,48]. In addition to the mCEQ, we asked participants to report their preferences regarding the IQOS^TM^ and e-cigarette. In general, the IQOS^TM^ was rated slightly higher than the e-cigarette on all preference scales, but only for “*wanting to use the product another five minutes*” there was a reliable difference.

All results obtained should be seen in the light of the following limitations. First, the results and interpretations of these results are based on a small convenience sample. It is possible that this sample is not representative for the average regular smoker. Almost all participants were students, relatively young, and not highly cigarette dependent. Second, participants only received a brief explanation on how to use the two products, which were new for them, and they got no more than five minutes to use each product as much or as little as they wanted. This short-term use can as well have had an impact on the results. In the studies of PMI, participants were five days in confinement, and needed to switch completely to the IQOS^TM^. In that case, participants got the opportunity to become more familiar with the product and to gain more knowledge on how to use the IQOS^TM^ in the most effective way. Finally, and probably most importantly, for both the HnB product and the e-cigarette, we used just one specific type. Whereas the offer of the different HnB products is rather limited, a plethora of types of e-cigarettes and of flavors of e-liquids, with various nicotine concentrations exists, which may well differ with respect to the effect on the behavioral and experiential variables studied here. Therefore, one should be very cautious extrapolating these results to other types of HnB products and most of all, to other types of e-cigarettes.

## 5. Conclusions

To conclude, within the framework of THR, in which smokers ideally should be able to freely choose from a variety of less harmful alternatives for smoking, HnB products seem to have the potential of a promising new offering. Our study namely showed that short-term use of a specific HnB product, IQOS^TM^, can be effective to momentarily reduce acute cigarette craving and withdrawal symptoms, while having a minimal impact on the eCO levels, and being slightly more liked by novice users than an e-cigarette. While these short-term effects are promising, they do of course not guarantee that craving/withdrawal reduction will also be sustained over longer time spans or in case of repeated use, nor do they provide assurance that these effects are sufficient to lead to smoking reduction or cessation in smokers willing to quit or cut down on cigarettes. Therefore, it will be important to further independently investigate the effects of HnB products in the long-term, not only with respect to health-effects, but also with respect to their potential as a (partial, or preferably, complete) substitute for smoking cigarettes.

## Figures and Tables

**Figure 1 ijerph-15-02902-f001:**
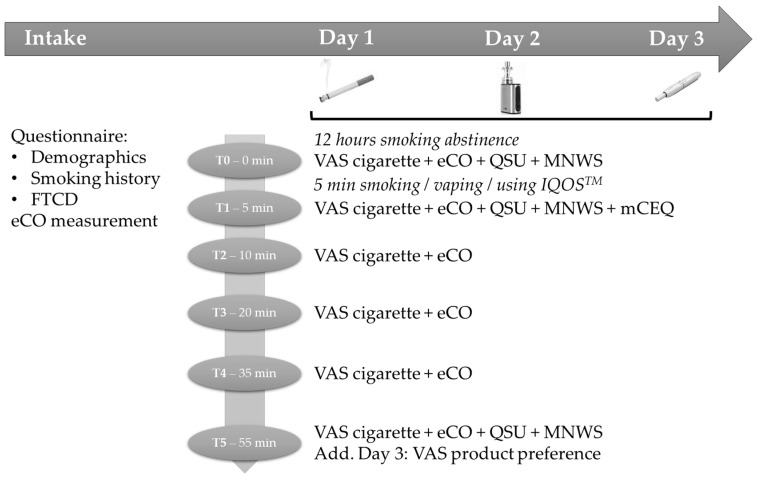
Study design and procedure.

**Figure 2 ijerph-15-02902-f002:**
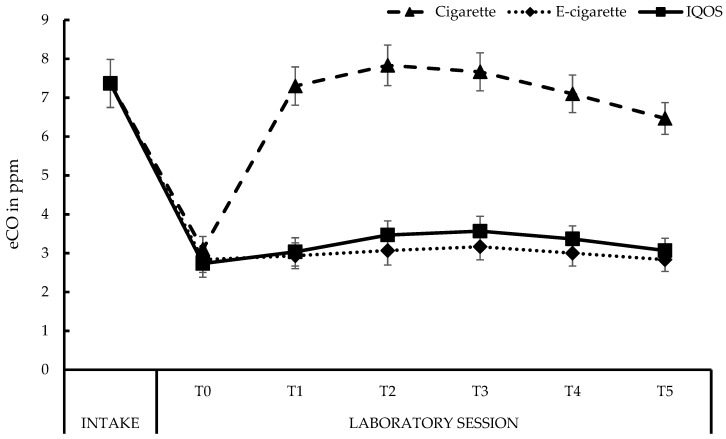
Exhaled CO (eCO) levels in ppm. **Cigarette**: *M*_Intake_ (*SE* between brackets) = 7.37 (0.62), *M*_T0_ = 3.10 (0.33), *M*_T1_ = 7.30 (0.49), *M*_T2_ = 7.83 (0.52), *M*_T3_ = 7.67 (0.49), *M*_T4_ = 7.10 (0.48), *M*_T5_ = 6.47 (0.41); **E-cigarette**: *M*_Intake_ = 7.37 (0.62), *M*_T0_ = 2.83 (0.33), *M*_T1_ = 2.93 (0.33), *M*_T2_ = 3.07 (0.37), *M*_T3_ = 3.17 (0.34), *M*_T4_ = 3.00 (0.33), *M*_T5_ = 2.83 (0.30); **IQOS^TM^**: *M*_Intake_ = 7.37 (0.62), *M*_T0_ = 2.73 (0.35), *M*_T1_ = 3.03 (0.36), *M*_T2_ = 3.47 (0.36), *M*_T3_ = 3.57 (0.38), *M*_T4_ = 3.37 (0.34), *M*_T5_ = 3.07 (0.32).

**Figure 3 ijerph-15-02902-f003:**
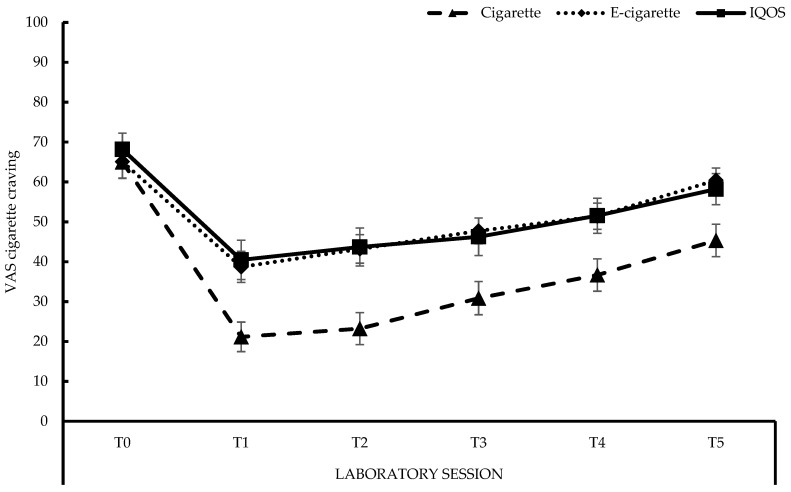
Visual Analogue Scale (VAS) cigarette craving. **Cigarette**: *M*_T0_ (*SE* between brackets) = 65.00 (4.13), *M*_T1_ = 21.17 (3.71), *M*_T2_ = 23.23 (4.01), *M*_T3_ = 30.87 (4.17), *M*_T4_ = 36.67 (4.06), *M*_T5_ = 45.33 (4.05); **E-cigarette**: *M*_T0_ = 65.07 (4.07), *M*_T1_ = 38.70 (3.88), *M*_T2_ = 43.20 (3.57), *M*_T3_ = 47.73 (3.21), *M*_T4_ = 51.40 (3.29), *M*_T5_ = 60.43 (3.05); **IQOS^TM^**: *M*_T0_ = 68.17 (4.08), *M*_T1_ = 40.47 (4.94), *M*_T2_ = 43.70 (4.77), *M*_T3_ = 46.27 (4.71), *M*_T4_ = 51.53 (4.40), *M*_T5_ = 58.20 (3.89).

**Figure 4 ijerph-15-02902-f004:**
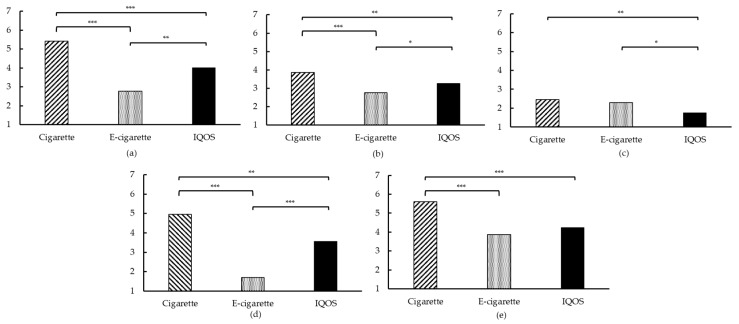
Modified Cigarette Evaluation Questionnaire (mCEQ) subscales. (**a**) “*Satisfaction*”; (**b**) “*Psychological reward*“; (**c**) “*Aversion*“; (**d**) “*Enjoyment of respiratory tract sensations*“; (**e**) “*Craving reduction*”; for (a–e): * *p* < 0.05, ** *p* < 0.01, *** *p* < 0.001.

**Figure 5 ijerph-15-02902-f005:**
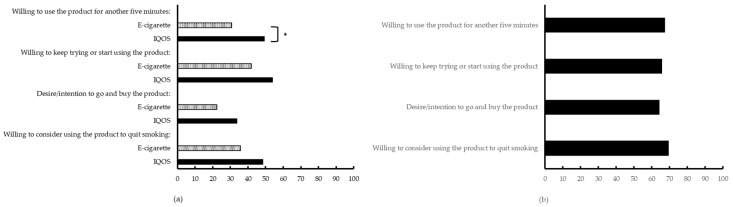
Additional VASs on Day 3. (**a**) Preferences for e-cigarette and IQOS^TM^, separately, with 0 being “*Not at all*” and 100 “*Very much so*”; (**b**) Product preference with 0 being “*E-cigarette*” and 100 being ”*IQOS*”; for both (a and b) * *p* < 0.05.

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
