# Peer review of "IQOSTM vs. e-Cigarette vs. Tobacco Cigarette: A Direct Comparison of Short-Term Effects after Overnight-Abstinence"

_ijerph, 2018, doi:10.3390/ijerph15122902_

Round 1
Reviewer 1 Report
1) Introduction is very long yet the first paragraph is very general and although includes references does not provide any specific figure, for example, it is mentioned that quit success is very low yet there is no exact number. Providing more concrete numbers is essential in setting the context for the reader.
2) References to original papers should be used rather than summary reports [Page 1 Line 39 onwards].
3) Page 2 Line 54-56 authors suggest a potential reason for dual use. However, there is no reference to where this suggestion is coming from. Is it their own idea? Or has been suggested in previous research?
4) Very superficial description of research on e-cigarettes has been provided in comparison to very detailed description of the few IQOS studies. For proper introduction balanced literature review should be provided as currently it appears that already before presenting study aims authors favour IQOS. The introduction could also be considerably shortened by reducing the level of detail of IQOS studies.
5) What was the justification for inclusion criteria, for example, smoking at least 10 cigarettes a day?
6) Why authors have set out inclusion and exclusion criteria, but when participants were recruited participants who did not meet inclusion criteria were still included in the sample? Why specifically these two participants were included in the study? (Page3 Line 142-144)
7) In the results section authors refer to tables in the appendix, for example, Tables B3 when referring to no differences in withdrawal symptoms yet the actual Table B3 is really unclear (as are other tables). Authors report intercept and error, but it is not possible to read the table without looking for the reference in the text. These tables should be presented in the way that a reader can understand the reported results easily without looking for detailed description in the text.
8) Figure 4- it is impossible to read figure labels- this should be corrected
9) One of the items for product evaluation was enjoyment of respiratory tract sensation- what do authors mean by that? Is it meant to represent how similar to smoking using other products is? Did participants receive an explanation on this?
10) Conclusions are disproportionally strong compared to the limitations mentioned in the discussion suggesting that HnB products are promising for harm reduction as the evidence presented only allow to draw very limited conclusions.
Author Response
First, we would like to thank the Reviewer for the extensive feedback and the useful suggestions regarding our manuscript. The feedback allowed us to take a closer look at some of the weaknesses of the submitted version of our manuscript. We believe that the adaptations made based on the feedback of the Reviewer have definitely improved the manuscript. Below we discuss how and where we have made changes based on the points raised:
1) Introduction is very long yet the first paragraph is very general and although includes references does not provide any specific figure, for example, it is mentioned that quit success is very low yet there is no exact number. Providing more concrete numbers is essential in setting the context for the reader.
We thank the Reviewer for this suggestion. We added more details concerning long-term quit-smoking rates when using willpower or smoking cessation aids, see lines 33-38.
2) References to original papers should be used rather than summary reports [Page 1 Line 39 onwards].
We thank the Reviewer for the recommendation. As we state in our manuscript, research within the domain of e-cigarette exists for about a decade and therefore we included references of summary reports, rather than individual, original papers. Those more interested in individual papers, can find the references in the summary reports. However, for the readers’ convenience, we also added some references to original papers to the sentences, see references 16-18 and 20-21.
3) Page 2 Line 54-56 authors suggest a potential reason for dual use. However, there is no reference to where this suggestion is coming from. Is it their own idea? Or has been suggested in previous research?
The results of the study described in the following sentence (“For example, …”) explains this potential reason for dual use and/or for relatively low uptake of vaping by smokers. We apologize that this was not clear and added the reference (28) to this sentence as well. We also added an additional reference concerning specific reasons for dual use, see reference 29:
· Robertson, L.; Hoek, J.; Blank, M-L.; Richards, R.; Ling, P.; Popova, L. Dual use of electronic nicotine delivery systems (ENDS) and smoked tobacco; A qualitative analysis. Tob Control 2018, 0, 1-7, doi:10.1136/tobaccocontrol-2017-0540.
4) Very superficial description of research on e-cigarettes has been provided in comparison to very detailed description of the few IQOS studies. For proper introduction balanced literature review should be provided as currently it appears that already before presenting study aims authors favour IQOS. The introduction could also be considerably shortened by reducing the level of detail of IQOS studies.
Unlike suggested by the Reviewer, our choice to report more extensively on research on IQOS than on research on e-cigarettes, has nothing to do with a preference for either product category. We simply do not favour IQOS, nor do we favour e-cigarettes. However, as described in suggestion number 2, we mainly reported summary statements concerning e-cigarettes because the e-cigarette literature contains thousands of papers published over the last decade that already have been critically evaluated and integrated in several excellent summary reports. This field is currently more elaborated and established, especially compared to the much younger and less developed research field of HnB products. Therefore we only refer to the main important summary reports concerning e-cigarettes (as a source of reference for the interested – maybe new to the field – reader), and have written more extensively on the existing literature concerning HnB products. A lot of research has been conducted by several tobacco companies, in contrast to little research by independent researchers. The main aim of the introduction was to highlight this imbalance between research from tobacco companies and from independent researchers. We focus on the specific literature that is of most importance for the main aim of the current study (namely, investigating the effect of using a specific HnB product (IQOS) on several psychological aspects such as cigarette craving and withdrawal symptoms). Therefore, we did not make any changes in this part of the introduction and we hope that this additional information is sufficient for the Reviewer.
5) What was the justification for inclusion criteria, for example, smoking at least 10 cigarettes a day?
The main reasons for using inclusion criteria were that we wanted to a) focus on the group of people to whom e-cigarettes and heat-not-burn products are mainly targeted (current smokers); b) to study (the effects of) product use in first-time users; c) and that we were mainly interested in the efficacy of both products with respect to cigarette craving reduction after a period of abstinence. We therefore used criteria to recruit only regular smokers, who were novel users for both e-cigarettes and HnB products, and who were smoking abstinent before each lab sessions. Specific for the criterion of smoking at least 10 CPD, we have based ourselves on our previous research from our group, as well as on research by others, for example:
· Adriaens, K.; Van Gucht, D.; Declerk, P.; Baeyens, F. Effectiveness of the electronic cigarette: an eight-week Flemish study with six-month follow-up on smoking reduction, craving and experienced benefits and complaints. IJERPH 2014, 11, 11220-11248, doi:10.3390/ijerph111111220.
· Caponnetto, P.; Maglia, M.; Prosperini, G.; Busà, B.; Polosa, R. Carbon monoxide levels after inhalation from new generation heated tobacco products. Respir Res 2018, 19, https://doi.org/10.1186/s12931-018-0867-z.
6) Why authors have set out inclusion and exclusion criteria, but when participants were recruited participants who did not meet inclusion criteria were still included in the sample? Why specifically these two participants were included in the study? (Page3 Line 142-144)
We apologize for the confusion about the inclusion criteria. First and most important for the central research questions of the current study, included participants needed to be regular smokers, novel to both e-cigarettes and HnB products and smoking abstinent before each lab session. Other inclusion criteria (e.g., no quit smoking motivation) were of less importance. One participant smoked on average 9 CPD (so probably some days around 10, and very close to the inclusion criteria), and met all other criteria. The second participant met all criteria, indicated in the Informed Consent no intention to quit smoking, but in the questionnaire answered “to maybe quit within the next three months”. By including both participants, we were able to have a balanced design. In addition, when conducting the main analyses (eCO, VAS cigarette craving, MNWS-R, mCEQ) without one or both participants, the main results remain the same, please see the ANOVA tables in the added excel file. We have therefore decided to keep both participants included, so that the design was kept balanced. We added clarification for the readers, see lines 146, 148-151.
7) In the results section authors refer to tables in the appendix, for example, Tables B3 when referring to no differences in withdrawal symptoms yet the actual Table B3 is really unclear (as are other tables). Authors report intercept and error, but it is not possible to read the table without looking for the reference in the text. These tables should be presented in the way that a reader can understand the reported results easily without looking for detailed description in the text.
We have included additional information so that the tables in Appendix B are more self-explanatory.
8) Figure 4- it is impossible to read figure labels- this should be corrected
We thank the reviewer for this suggestion. The font sizes of the figure labels of Figure 4 have been enlarged to improve readability.
9) One of the items for product evaluation was enjoyment of respiratory tract sensation- what do authors mean by that? Is it meant to represent how similar to smoking using other products is? Did participants receive an explanation on this?
In Figure 4 all subscales of the questionnaire assessing product evaluation are displayed, with one of them being “Enjoyment of respiratory tract sensation”. This subscale included only a single-item question, which was displayed to participants as follows: “Did you enjoy the sensations in your throat and chest?”. The item itself is indeed more self-explanatory than the rather vague or abstract name of the subscale. As described in section 2.4.2. “Subjective effect questionnaires”, more information can be found in the following reference: Cappelleri, J.C.; Bushmakin, A.G.; Baker, C.L.; Merikle, E.; Olufade, A.O.; Gilbert, D.G. Confirmatory factor analyses and reliability of the modified cigarette evaluation questionnaire. Addict Behav 2007, 32, 912-923, doi:10.1016/j.addbeh.2006.06.028.
10) Conclusions are disproportionally strong compared to the limitations mentioned in the discussion suggesting that HnB products are promising for harm reduction as the evidence presented only allow to draw very limited conclusions.
We thank the Reviewer for this suggestion. We have made some adjustments in the Conclusions section so that the conclusions are more balanced, See lines 480-482, 484-489.
Reviewer 2 Report
This is a well-written manuscript. The introduction presents existing knowledge related to the topic, identifies gaps in the literature, and clearly states the aims of the study. The design of the study is described and the results are clearly presented. I believe this research would be of interest to clinicians seeking a tobacco harm reduction approach. My only suggestion relates to the statement regarding the inclusion of two participants who did not meet inclusion criteria (lines 142-143). I think it would be appropriate to provide readers with an explanation of the rationale for including these participants in the study.
Author Response
First of all we would like to thank the Reviewer for his/her praising words concerning our manuscript. We apologize for the confusion about the inclusion criteria. First and most important for the central research questions of the current study, included participants needed to be regular smokers, novel to both e-cigarettes and HnB products and smoking abstinent before each lab session. Other inclusion criteria (e.g., no quit smoking motivation) were of less importance. One participant smoked on average 9 CPD (so probably some days around 10, and very close to the inclusion criteria), and met all other criteria. The second participant met all criteria, indicated in the Informed Consent no intention to quit smoking, but in the questionnaire answered “to maybe quit within the next three months”. By including both participants, we were able to have a balanced design. In addition, when conducting the main analyses (eCO, VAS cigarette craving, MNWS-R, mCEQ) without one or both participants, the main results remain the same, please see the ANOVA tables in the added excel file. We have therefore decided to keep both participants included, so that the design was kept balanced. We added clarification for the readers, see lines 146, 148-151.
Round 2
Reviewer 1 Report
Although generally I think the comments have been addressed I do think that the introduction is still excessively long and does not present balanced view (particularly as authors state that they do not support one product more than the other). Also, inclusion criteria are justified and explained in the response document but if these are based on previous research it should be stated in methods with appropriate references.
Author Response
We thank the Reviewer for the second round of revision.Although we still believe in the validity of the arguments underlying our choices made concerning the introduction (see also first Review round), we have added additional information concerning quit-smoking rates using e-cigarettes, see lines 50-56. By doing so, we hope that we have sufficiently met the suggestion of the Reviewer.
We thank the reviewer for the suggestion concerning the justification of our inclusion criteria and we made the necessary adjustments by adding some references, see line 144.